# OUT-OF-DISTRIBUTION DETECTION USING NEURAL ACTIVATION PRIOR

## ABSTRACT

Out-of-distribution detection (OOD) is a crucial technique for deploying machine learning models in the real world to handle the unseen scenarios. Compared to standard classification tasks, OOD detection presents significant challenges due to the unpredictable nature and inherent difficulty in collecting OOD data. Consequently, a natural solution is to develop priors that are as diverse as possible, effectively characterizing the features of OOD data. In this paper, we first propose a simple yet effective **N**eural **A**ctivation **P**rior (NAP) for OOD detection. Our prior is based on a key observation that, for a channel before the pooling layer of a fully trained neural network, the probability of a few neurons being activated with a large response by an in-distribution (ID) sample is significantly higher than that by an OOD sample. An intuitive explanation is that for a model fully trained on ID dataset, each channel would play a role in detecting a certain pattern in the ID dataset, and a few neurons can be activated with a large response when the pattern is detected in an input sample. Then, an effective scoring function based on this prior is proposed to highlight the role of these strongly activated neurons in OOD detection. Our approach is plug-and-play and does not lead to any performance degradation on ID data classification and requires no extra training or statistics from training or external datasets. To the best of our knowledge, our method is the first to exploit intra-channel activation pattern information, contributing to its orthogonality to existing approaches and allowing it to be effectively combined with them in various applications. Furthermore, we conduct an elegant oracle experiment to validate the rationale behind our proposed scoring function. Extensive experimental results demonstrate the effectiveness of our method. Moreover, our approach can significantly boost the performance when integrated with most existing methods, showcasing the unique attributes of the proposed prior.

## 1 INTRODUCTION

Deep learning has developed rapidly in the last decade and become a crucial technique in various fields. However, neural networks would frequently make erroneous judgments in inference when encountering the data that differs greatly from their training data, which is known as out-of-distribution (OOD) data. This challenge is particularly vital in safety-critical areas such as autonomous driving (Filos et al., 2020; Janai et al., 2020) and medical diagnosis (Pooch et al., 2020), which urges the development of effective OOD detection methods.

In practice, OOD data exhibits large diversity and is difficult to identify (Yang et al., 2021). Existing studies typically formulate OOD detection as a one-class classification task, utilizing prior knowledge. They (Hendrycks & Gimpel, 2016; Liu et al., 2020; Yu et al., 2023) propose various priors, based on which they further design scoring functions to distinguish OOD samples from ID samples. For example, Hendrycks & Gimpel (2016) observed that OOD samples always exhibit lower maximum softmax probabilities, and accordingly proposed using the maximal softmax probability output by a neural network as an OOD indicator. Liu et al. (2020) found that OOD samples usually have lower logits values, and based on this, an energy function was proposed for OOD detection. Drawing from these precedents, it's clear that existing methods largely rely on the introduction of certain priors. While some promising results highlight the effectiveness of these heuristics, a gap remains in meeting the practical requirements of real-world applications. Furthermore, some existing methods (Olber et al., 2023; Dong et al., 2022; Djurisic et al., 2022; Sun et al., 2021; Liu et al., 2023; Sun et al., 2022)

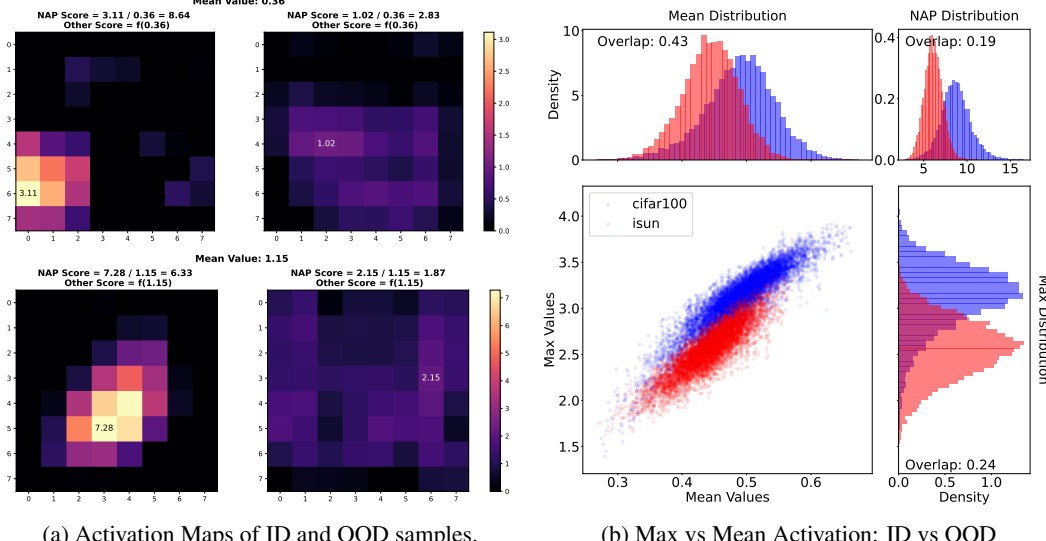

(a) Activation Maps of ID and OOD samples.          (b) Max vs Mean Activation: ID vs OOD

Figure 1: **Ignoring the distributional characteristics of activation values within channels makes it difficult to distinguish between ID and OOD samples.** (a) illustrates activation value maps in the penultimate layer for ID samples (left) and OOD samples (right) within the same channel (each row represents an example). Given identical mean channel activation values, ID samples exhibit high activation values within the channel (left) due to the detection of specific patterns, whereas OOD samples display a large number of mild, noise-like activations (right) resulting from the model's lack of training on OOD data. This phenomenon makes it impossible to distinguish between ID and OOD samples if we discard the characteristics within the channel while solely depending on mean activation values. More examples are provided in Appendix J. (b) depicts the activation distribution differences between ID and OOD data within the channel. Through the scatter plot (bottom-left), we observe that at the same average activation level, ID data (CIFAR-10) exhibits significantly higher maximum activation values than OOD data. In the top-left and bottom-right, we show the distributions of mean and max values respectively, highlighting the overlapping areas between ID and OOD sample distributions. In the top-right corner, we present the score distribution of our proposed NAP (which considers information from both mean and max dimensions). It is evident that by considering both dimensions simultaneously, the degree of overlap between ID and OOD distributions is substantially reduced. Furthermore, when combined with other methods (as demonstrated in Fig 2 ), we can further minimize the overlapping area between these distributions.

have established that activations in the penultimate layer contain rich information, which is proved to be highly valuable for detecting OOD samples. However, we observe that the feature characteristics of the penultimate layer prior to pooling have been consistently overlooked. We posit that identifying and incorporating priors that can complement these existing approaches is essential. This insight forms the foundation of our work and constitutes its primary contribution.

In this paper, we propose a novel prior, called **N**eural **A**ctivation **P**rior (NAP), for OOD detection. NAP characterizes our key observation that, in fully trained neural networks, certain neurons in channels *preceding the global pooling layer (Figure 3)* exhibit significantly higher activation probabilities for ID samples compared to OOD samples. The rationale behind NAP is that channels in models trained on ID datasets develop sensitivity to specific patterns within that distribution. When these patterns are detected, a subset of neurons activates strongly (Hoefler et al., 2021). Such pronounced activations are predominantly observed with ID inputs and are rare for OOD data, which typically lack these learned patterns. To validate NAP, we analyzed the mean and maximum within-channel activations at the penultimate layer of DenseNet (Huang et al., 2017), trained on CIFAR-10 (Krizhevsky et al., 2009) and iSun (Xu et al., 2015) datasets (Figure 1). It clearly demonstrates that ID samples exhibit significantly higher maximal activation values than OOD samples at equal average activation levels. Consequently, existing methods based on pooled activation values struggle to effectively distinguish these OOD samples, given the non-discriminative nature of their average activation values.

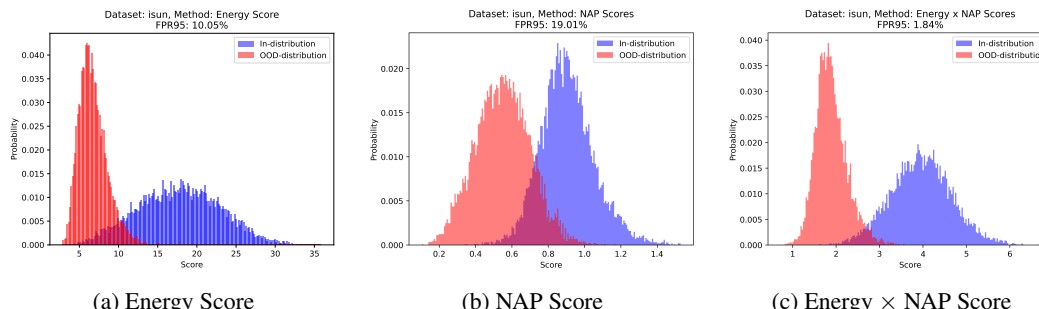

(a) Energy Score        (b) NAP Score        (c) Energy × NAP Score

Figure 2: **Score distribution visualization using DenseNet on CIFAR-10 (ID) and iSun (OOD)** The integration of (a) Energy Score and (b) NAP Score through multiplication yields the (c) Energy × NAP Score, demonstrating superior differentiation between ID and OOD datasets. The effectiveness of this approach is attributed to the orthogonal nature of the proposed NAP relative to conventional OOD detection methods exemplified by the Energy Score. This illustrates that a simple multiplicative combination with NAP enhances detection capability. Importantly, the objective is not merely to surpass the performance of the Energy Score itself, but to underscore the synergistic potential of NAP as a complementary enhancement to the Energy Score and similar method.

We argue that the prior proposed in this work is orthogonal to existing methods, offering a novel perspective in OOD detection. As evidenced by the third column of Table 1, current activation-based OOD detection techniques have overlooked the valuable information contained in intra-channel distribution patterns. Our prior effectively addresses this gap in the literature. Consequently, the primary contribution of this research lies not in direct comparison with existing approaches, but in demonstrating the substantial performance enhancements achieved by integrating our proposed prior with state-of-the-art techniques.

Furthermore, we introduce a novel and effective scoring function for OOD detection based on our prior NAP. To be precise, NAP utilizes the ratio of maximal to averaged activation values within a channel. The rationale behind this scoring function can be understood from both conceptual and empirical perspectives. Conceptually, inspired by the Signal-to-Noise Ratio (SNR), we consider the maximal activation value as signal strength and the averaged value as noise. Consequently, their ratio serves as a quantitative measure of the channel's information fidelity. For a comprehensive discussion, refer to Section 4.2. Empirically, Figure 1 provides compelling evidence that this ratio exhibits substantially higher values for in-distribution samples compared to out-of-distribution data. In practical deployment, our scoring function complements existing metrics, enhancing OOD detection when combined multiplicatively (Figure 2). Also, it is noteworthy that this scoring function is a plug-and-play method, requiring no additional training, extra data, or reliance on pre-calculated statistical data from the training set, which makes it broadly applicable. A more detailed comparative analysis is provided in Table 1, elucidating the distinctive features of our approach in relation to existing methods.

Our experimental results not only validate the effectiveness of our proposed prior but also highlight its substantial contribution to the OOD detection field. Specifically, our prior enriches the community's library of priors for OOD detection due to its orthogonality to previous ones. This unique characteristic allows it to enhance detection accuracy across a variety of existing methods. Our findings indicate that when used independently, our OOD detection method achieves performance comparable to current state-of-the-art techniques across multiple datasets and architectures. Moreover, integrating our approach with existing ones leads to significant improvements, with a reduction in FPR95 by up to 66.03%.

In summary, our contributions are as follows:

- We introduce the Neural Activation Prior (NAP), a novel contribution to OOD detection. Uniquely, NAP is orthogonal to priors utilized in existing methods, offering a distinct and complementary perspective that paves the way for advanced OOD detection research.

- Based on the proposed prior, we develop a simple yet effective OOD detection scoring function. It is a plug-and-play approach that does not impair the model's inherent capabilities.

It can be readily integrated with many existing OOD detection techniques, enhancing their ability to balance OOD detection with ID accuracy.

- We demonstrate the state-of-the-art performance of our approach through extensive experiments across various datasets, including a reduction in FPR95 by up to 66.03%. These results underscore the method's operational efficiency, simplicity of deployment, and overall efficacy.

## 2 RELATED WORK

### 2.1 OOD DETECTION

The OOD detection community has explored a variety of techniques to underscore the distinctions between ID and OOD samples. These methods encompass classification-based (Huang & Li, 2021; Liang et al., 2018; Bendale & Boult, 2016; DeVries & Taylor, 2018; Hendrycks & Gimpel, 2016; Sastry & Oore, 2020; Tack et al., 2020; Du et al., 2022), density-based (Zong et al., 2018; Abati et al., 2019; Nalisnick et al., 2018; Zisselman & Tamar, 2020; Jiang et al., 2021; Pidhorskyi et al., 2018; Kirichenko et al., 2020; Sabokrou et al., 2018), and distance-based approaches (Lee et al., 2018b; Ming et al., 2022; Chen et al., 2020; Zaeemzadeh et al., 2021; Van Amersfoort et al., 2020; Techapanurak et al., 2020; Lu et al., 2023; Sun et al., 2022), with classification-based techniques generally outperforming the other types (Yang et al., 2021). In classification-based methods, the basic work of OOD detection starts with a simple and effective baseline: using the Maximum Softmax Probability (MSP) (Hendrycks & Gimpel, 2016) to measure the probability that a certain sample is an ID sample. On this basis, early approaches (Liang et al., 2018; Hsu et al., 2020; Liu et al., 2020) focused on developing enhanced OOD indicators derived from neural network outputs. In addition, some researchers have proposed strategies involving OOD sample generation (Lee et al., 2018a; Du et al., 2022) and gradient-based (Liang et al., 2018) techniques. Among these, certain post-hoc methods (Hendrycks & Gimpel, 2016; Liang et al., 2018; Liu et al., 2020; Sun et al., 2021; Sun & Li, 2022; Djurisic et al., 2022; Yu et al., 2023; Du et al., 2022) are notable for their simplicity and because they do not necessitate changes in the training process or objectives. This feature is particularly valuable for implementing OOD detection in real production environments, where the additional cost and complexity associated with retraining would be unacceptable.

The MSP method, initially presented by Hendrycks & Gimpel (2016), was a formative step in post hoc OOD detection, using a neural network's softmax output as a heuristic for distinguishing ID from OOD samples. Its straightforward application facilitated early adoption in OOD studies. Despite MSP's influence, its limitations prompted further innovation, giving rise to the Energy method. This method, proposed by Liu et al. (2020), refines the approach by assigning an energy score to network outputs, showing quantitative improvements over MSP with theoretical and empirical support. Advancements in post hoc OOD detection have led to diverse methodological branches stemming from MSP and Energy paradigms. Ahn et al. (2023) innovates by reducing neuron-induced noise through the calculation of Shapley values. Yu et al. (2023) distinguish ID from OOD data by identifying neural network blocks with optimal differentiation based on the norms of their features. Sun & Li (2022) improves discrimination by pruning weights in the fully connected layer according to the contribution units make during classification. On the other end of the spectrum, entirely computation-free post hoc methods such as ReAct (Sun et al., 2021) and ASH (Djurisic et al., 2022) have shown promise. Sun et al. (2021) investigates activations prior to the fully connected layer, applying rectification to suppress extreme activations that OOD data tend to trigger, thereby achieving refined detection outcomes. Similarly, Djurisic et al. (2022) prunes the activations inputted to the fully connected layer, but it achieves even more enhanced results compared to Sun & Li (2022) by its selective pruning strategy. In this paper, our comparison mainly focuses on post hoc methods, since our method also belongs to this category.

Recent advancements in OOD detection have highlighted the rich informational content embedded within the penultimate layer activations. Notable contributions in this domain include NAPattern (Olber et al., 2023), NMD (Dong et al., 2022), ASH (Djurisic et al., 2022), ReAct (Sun et al., 2021), NAC-UE (Liu et al., 2023), and KNN (Sun et al., 2022). Table 1 presents a comprehensive analysis of these methods, juxtaposing them with our proposed NAP approach. Our method's distinctive feature lies in its exploitation of intra-channel activation patterns, a dimension largely unexplored by existing techniques. This novel perspective renders NAP complementary to current methodologies.

Table 1: **The orthogonality of NAP to existing methods related to the penultimate layer.** The orthogonality of NAP to methods not listed in the table is evident, as these other methods utilize other the feature of neural network like logits and confidence scores. In contrast, our method captures this crucial information. No Thr.: No threshold (for data preprocessing) tuning required. No Train: No additional training needed. Comp.: No additional computation needed. Stor.: No additional storage needed. Intra: Use of intra-channel activation patterns. Inter: Use of inter-channel activation patterns. #Param: Number of hyperparameters requiring tuning. Note: #Params of NMD > 4 due to its need for additional training.

| Method | Prior design based on Penultimate Layer Activations | No Thr. | No Train | Comp. | STor. | Intra | Inter | #Param |
|---|---|---|---|---|---|---|---|---|
| NAPattern (Olber et al., 2023) | Binarize the activation values after pooling using a given threshold. Perform OOD detection by calculating the Hamming distance between this binary vector and each binary vector in training set. | ✗ | ✓ | ✗ | ✗ | ✗ | ✓ | 4 |
| NMD (Dong et al., 2022) | Train a binary classifier on pooled activations from ID and OOD samples. | ✓ | ✗ | ✗ | ✗ | ✗ | ✓ | > 4 |
| ASH (Djurisic et al., 2022) | Apply pruning and exponential weighting to the pooled activation vector to amplify significant values. | ✗ | ✓ | ✓ | ✓ | ✗ | ✓ | 1 |
| ReAct (Sun et al., 2021) | Rectify the larger activation values after pooling. | ✗ | ✓ | ✓ | ✓ | ✗ | ✓ | 1 |
| NAC-UE (Liu et al., 2023) | Compute the distribution of pooled activation values' contributions to deviations from uniform output using gradient computation, and use this distribution to set a threshold for OOD detection. | ✗ | ✓ | ✗ | ✓ | ✗ | ✓ | 3 |
| KNN (Sun et al., 2022) | Compute the nearest neighbor distance between a test sample and training samples in the pooled activation space to determine if the sample is out-of-distribution | ✓ | ✓ | ✗ | ✗ | ✗ | ✓ | 2 |
| NAP (Ours) | Capture activation patterns both intra-channel and inter-channel before pooling, and utilize the combination of these patterns for OOD detection. | ✓ | ✓ | ✓ | ✓ | ✓ | ✓ | 1 |

As evidenced by the empirical results presented in Tables 2 and 3, the integration of our approach with existing methods yields substantial performance enhancements. Crucially, this performance boost comes at a negligible computational cost. The overhead introduced by incorporating NAP is a mere 0.0462% increase in processing time, as detailed in Section 6.5. This minimal computational burden, coupled with significant performance gains, underscores the efficiency and effectiveness of our proposed method.

## 3 NEURAL ACTIVATION PRIOR

Our NAP is based on the following observation for OOD detection: for a channel located before the global pooling layer in a fully trained neural network, the likelihood that a small number of its neurons activated with a stronger response to an ID sample is significantly higher compared to an OOD sample. For the behavior in other layers of the neural network, refer to the discussion in Appendix C.

To formally describe this observation, we first define the concept of neural activation. Consider a trained classification neural network $f$, assuming it receives $D$-dimensional input $x$ and outputs $K$-dimensional logits. That is $f : \mathbb{R}^D \to \mathbb{R}^K$. We concentrate on the activation tensor $\mathbf{A}$ , located at the penultimate layer just before the global pooling operation, as illustrated in Figure 3. Let the dimensions of $\mathbf{A}$ be $C \times H \times W$, where $C$ is the number of channels, and $H$ and $W$ are the spatial dimensions.

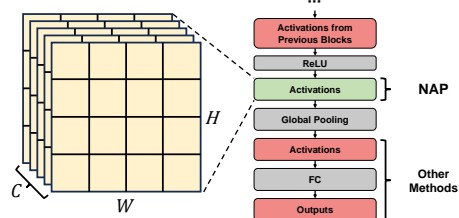

Figure 3: **Illustration identifying the focus zone of NAP in neural networks.** The figure highlights the specific position of NAP in the network.

Let $\mathbf{A}_j$ represent the activation tensor of the $j$-th channel. We define two key statistical indicators of $\mathbf{A}_j$ as follows:

- **Maximum activation value:**

$$\text{Max}(\mathbf{A}_j) = \max_{k,l} \mathbf{A}_{jkl}, \tag{1}$$

  where $\text{Max}(\mathbf{A}_j)$ is the maximum value among all elements in the activation vector $\mathbf{A}_j$. Here, $k$ and $l$ index the spatial dimensions (height and width) of the activation map, respectively. Notice that $\mathbf{A}_{jkl}$ is processed by ReLU (as shown in Figure 3), hence $\mathbf{A}_{jkl} \geq 0$.

- **Mean activation value:**

$$\text{Mean}(\mathbf{A}_j) = \frac{1}{h \times w} \sum_{k=1}^{h} \sum_{l=1}^{w} \mathbf{A}_{jkl}, \tag{2}$$

  where $\text{Mean}(\mathbf{A}_j)$ is the average of all activation values in the $j$-th channel.

Inspired by the concept of signal-to-noise ratio, we interpret $\text{Max}(\mathbf{A}_j)$ as the signal strength and $\text{Mean}(\mathbf{A}_j)$ as the noise strength. (for more detail discussion, refer to Section 4.2). Note that we calculate this metric separately for each channel in the network, as different channels are typically used to detect distinct patterns. The scatter plot of the mean and max values is depicted in Figure 1. It shows that for the comparable $\text{Mean}(\mathbf{A}_j)$ values, the $\text{Max}(\mathbf{A}_j)$ of ID samples is significantly greater than that of OOD samples. Therefor, it is natural to design a score function for OOD detection based on this observation, which we will discuss in detail in Section 4.

It is worth noting that our proposed prior is orthogonal to existing OOD detection methods. As illustrated in Table 1 and Figure 3, existing methods mainly focus on the network output and weight of the penultimate layer after global pooling, and leverage them to design various scoring functions for OOD detection. In contrast, the prior we propose focuses on the channels of the penultimate layer before global pooling. Since the distribution information within these channels is inevitably lost during the global pooling process, our proposed prior is complementary to existing work.

## 4 OOD DETECTION WITH NAP

### 4.1 BASICS

First, we will provide a brief overview of typical settings for OOD detection in image classification networks. Typically, classification networks are trained on ID data, that is, known training data sets. Once training is completed, the model can effectively classify the categories within the training data set. During the inference, samples mixed with OOD data are fed into this trained model. To identify OOD samples, researchers usually introduce a scoring function into the model. The model not only classifies each sample, but also uses a scoring function to generate a score for each one. This score is used to predict whether a sample belongs to an ID class in the training set, or an unknown OOD class.

### 4.2 SCORING FUNCTION DESIGN

Based on our prior proposed previously in Section 3, we propose a SNR-like scoring function. In our formulation, the mean activation value is interpreted as the noise intensity, while the maximal activation value is regarded as the signal strength. This conceptual framework leads to the following scoring function:

$$S_{NAP}(x; f) = \frac{1}{C} \sum_{j=1}^{C} \left( \frac{\text{Max}(\mathbf{A}_j)}{\text{Mean}(\mathbf{A}_j) + \epsilon} \right)^2, \tag{3}$$

where $C$ represents the number of channels before pooling. Note that a small constant $\epsilon > 0$ is added to ensure the numerical stability of the computation. Here, the square operator is used to adaptively emphasize the more significant channels, with an empirical analysis provided in Section 6.4. The scoring function $S_{NAP}(x; f)$ used in this paper is analogous to existing methods, wherein ID data is assigned a higher score, and OOD data is given a lower score.

**[Rationale behind the scoring function design.]** We discuss the reasoning behind our scoring function design to clarify the use of activation values as follows:

**I. *Why maximum value represent signal strength?*** Each channel typically detects a specific pattern, and the area of the pattern in the image determines the size of the high activation region. The size of this region does not reflect whether the sample is an OOD sample. Therefore, using the mean to measure signal strength is unreasonable because it is influenced by the size of the pattern's area. Instead, the maximum activation value reflects the confidence in the detected pattern. Thus, we use the maximum activation value to indicate the activation intensity of the pattern.

**II. *Why mean value represent noise strength?*** Although variance is generally used to measure noise strength, the sparsity of activation values within the channel is influenced by the pattern's area size in the image, thus the size of the activation region largely determines the variance. Therefore, using variance to measure noise is unreasonable. Instead, dividing the maximum value by the mean can serve as a normalization term, removing the influence of activation values from non-pattern regions (also known as background noise). Thus, we use the mean as the noise term in this SNR-like format.

To substantiate the effectiveness of our scoring function design, we employ an "Oracle" experiment in Section 6.1, providing empirical evidence for its validity in various scenarios.

**Remark 1.** The scoring function we proposed is a **plug-and-play** approach that can be easily integrated into existing neural network architectures. It requires **no additional training** or **external data** and retains the model's inherent classification capabilities. These properties make it practical and suitable for a variety of applications.

**Remark 2.** While CNNs are commonly used in OOD detection tasks, Transformer architectures (Vaswani et al., 2017) have proven effective across various applications. Motivated by this, we extend our method for compatibility with Transformers. Details of our experimental validation, which confirm the method's robustness and adaptability to different architectures, are provided in the Appendix K.

### 4.3 COMBINED WITH EXISTING METHODS

Based on our earlier discussion of orthogonality, our method can be flexibly integrated with existing approaches. In this paper, we present a natural paradigm as follows for combining it with existing methods and leave the exploration of more optimal integration strategies for future work:

$$S_{NAP\text{-}Other}(x; f, w) = S_{\text{Other}}(x; f)^w \cdot S_{NAP}(x; f)^{1-w},$$

where $S_{\text{Other}}(x; f)$ represents the score from other methods, and $w$ is the weighting parameter. The parameter $w$ is constrained to the interval $(0, 1)$, and in our experiments, it typically takes values in $\{0.2, 0.3, \ldots, 0.8\}$. In the following sections, we propose a method to find the optimal $w$, and in Section 6.2, we conduct an analysis of $w$'s sensitivity, finding that the performance remains stable within the range $[0.2, 0.8]$. Also, we conducted experiments using $S_{NAP}$ independently and found that it achieves excellent OOD detection performance on its own. For a more detailed discussion, please refer to Section 6.3.

**[How to find a optimal parameter $w$?]** While our method demonstrates robustness to variations in the parameter $w$ (see Section 6.2), determining the optimal $w$ can be achieved using the following approach. We utilized a set of data transformation techniques (such as Gaussian noise, glass blur, motion blur, etc., more details in the Appendix H) to generate a corrupted dataset based on the ID dataset, serving as pseudo OOD data. For the choice of transformation types, we referred to Hendrycks & Dietterich (2018). Utilizing this set of crafted OOD data, we employed a binary search method to find the optimal $w$. Through experimentation with various datasets and methods, we found that this search approach quickly identifies the optimal $w$, which generalizes well to real OOD datasets. Please refer to Appendix H for more details about this process.

## 5 EXPERIMENTS

In this section, we conduct experiments on various real-world datasets. In our experiments, we use **NAP-[initial]** to denote the combination of NAP with another method, where **[initial]** represents the initial letter of the method's name (e.g., NAP-A for the combination with ASH). Specifically, we combine NAP with a series of common OOD detection methods, including MSP (Hendrycks

Table 2: **Comparison with competitive post-hoc OOD detection method on CIFAR benchmarks.** All values are percentages and are averaged over 6 OOD test datasets. Note: A. = Area Under the ROC Curve; F. = False Positive Rate at 95% True Positive Rate. Methods include MSP (Hendrycks & Gimpel, 2016), Energy (Liu et al., 2020), ASH (Djurisic et al., 2022), DICE (Sun & Li, 2022), SCALE (Xu et al., 2023), ReAct (Sun et al., 2021), and KNN (Sun et al., 2022).

| Method | | MSP | NAP-M | Energy | NAP-E | ASH | NAP-A | DICE | NAP-D | SCALE | NAP-S | ReAct | NAP-R | KNN | NAP-K |
|---|---|---|---|---|---|---|---|---|---|---|---|---|---|---|---|
| CIFAR-10 | F. ↓ | 48.69 | **19.09** | 26.55 | **9.02** | 15.05 | **11.14** | 20.83 | **11.66** | 12.26 | **9.26** | 26.45 | **9.18** | 16.12 | **7.79** |
| | A. ↑ | 92.52 | **95.11** | 94.67 | **98.15** | 96.91 | **97.48** | 95.24 | **97.47** | 97.27 | **98.00** | 94.67 | **98.02** | 96.79 | **98.38** |
| CIFAR-100 | F. ↓ | 80.13 | **48.20** | 68.45 | **32.61** | 41.40 | **35.40** | 49.72 | **32.34** | 32.40 | **27.35** | 62.27 | **25.71** | 44.91 | **33.63** |
| | A. ↑ | 74.36 | **88.45** | 81.19 | **92.84** | 90.02 | **91.21** | 87.23 | **92.23** | 90.58 | **91.17** | 84.47 | **93.18** | 86.58 | **91.54** |

Table 3: **Comparison with competitive post-hoc OOD detection method on ImageNet-1k Deng et al. (2009).** All values are percentages. Methods include MSP (Hendrycks & Gimpel, 2016), Energy (Liu et al., 2020), ASH (Djurisic et al., 2022), DICE (Sun & Li, 2022), SCALE (Xu et al., 2023), ReAct (Sun et al., 2021), and KNN (Sun et al., 2022).

| Method | OOD Datasets | | | | | | | | Average | |
|---|---|---|---|---|---|---|---|---|---|---|
| | iNaturalist | | SUN | | Places | | Textures | | | |
| | FPR95 ↓ | AUROC ↑ | FPR95 ↓ | AUROC ↑ | FPR95 ↓ | AUROC ↑ | FPR95 ↓ | AUROC ↑ | FPR95 ↓ | AUROC ↑ |
| MSP | 64.29 | 85.32 | 77.02 | 77.10 | 79.23 | 76.27 | 73.51 | 77.30 | 73.51 | 79.00 |
| **NAP-M** | **35.47** | **92.53** | **51.19** | **86.51** | **63.77** | **80.61** | **15.14** | **97.09** | **41.39** | **89.19** |
| Energy | 59.50 | 88.91 | 62.65 | 84.50 | 69.37 | 81.19 | 58.05 | 85.03 | 62.39 | 84.91 |
| **NAP-E** | **29.90** | **94.47** | **39.69** | **90.46** | **55.17** | **85.15** | **11.74** | **97.28** | **34.12** | **91.84** |
| ASH | 31.46 | 94.28 | 38.45 | 91.61 | 51.80 | 87.56 | 20.92 | 95.07 | 35.66 | 92.13 |
| **NAP-A** | **26.26** | **95.10** | **32.89** | **92.77** | **48.69** | **87.92** | **11.60** | **97.32** | **29.86** | **93.28** |
| DICE | 43.09 | 90.83 | 38.69 | 90.46 | 53.11 | 85.81 | 32.80 | 91.30 | 41.92 | 89.60 |
| **NAP-D** | **27.48** | **94.13** | **36.14** | **90.66** | **51.84** | **85.03** | **9.02** | **97.92** | **31.12** | **91.94** |
| SCALE | 24.67 | 94.45 | 36.53 | 91.64 | 48.38 | 87.33 | 18.25 | 96.65 | 31.96 | 92.52 |
| **NAP-S** | **20.83** | **95.60** | **37.14** | **92.07** | **50.69** | **87.36** | **10.68** | **97.93** | **29.84** | **93.24** |
| ReAct | 42.40 | 91.53 | 47.69 | 88.16 | 51.56 | 86.64 | 38.42 | 91.53 | 45.02 | 89.47 |
| **NAP-R** | **24.58** | **95.55** | **38.47** | **91.12** | **53.32** | **86.24** | **9.57** | **97.60** | **31.49** | **92.63** |
| KNN | 85.91 | 72.67 | 90.49 | 65.39 | 93.18 | 60.08 | 14.08 | 96.98 | 70.92 | 73.78 |
| **NAP-K** | **38.23** | **89.80** | **56.55** | **80.01** | **70.89** | **71.35** | **7.02** | **98.39** | **43.17** | **84.89** |

& Gimpel, 2016), Energy (Liu et al., 2020), ASH (Djurisic et al., 2022), DICE (Sun & Li, 2022), SCALE (Xu et al., 2023), ReAct (Sun et al., 2021), and KNN (Sun et al., 2022), denoted as NAP-M, NAP-E, NAP-A, NAP-D, NAP-S, NAP-R, and NAP-K, respectively. CIFAR-10, CIFAR-100, and ImageNet are used as ID datasets. For each combination of NAP with other methods, we use the approach described in Section 4.3 to determine the optimal combination parameter $w$. Detailed optimal $w$ values for different experimental setups can be found in Appendix H. It's noted that the value of $\epsilon$ in our scoring functions is consistently set to 1.0 for numerical stability. All experiments were conducted on an NVIDIA GeForce RTX 3090 GPU.

## 5.1 EVALUATION ON CIFAR BENCHMARKS

**Implementation details.** In our experiments, consistent with recent studies (Sun et al., 2021; Sun & Li, 2022; Djurisic et al., 2022), we utilized 10,000 test images from both CIFAR-10 (Krizhevsky et al., 2009) and CIFAR-100 (Krizhevsky et al., 2009) as ID data. To gauge the performance of the model, six widely-used OOD datasets were employed as benchmarks. These datasets include SVHN (Netzer et al., 2011), Textures (Cimpoi et al., 2014), iSUN (Xu et al., 2015), LSUN-Crop (Yu et al., 2015), LSUN-Resize (Yu et al., 2015), and Places365 (Zhou et al., 2017). As for pre-trained model, we employed DenseNet (Huang et al., 2017), and we follow the training setting of DenseNet introduced in Sun & Li (2022).

**Experimental results.** Table 2 presents the comparison of NAP combined with other post hoc OOD detection methods on the CIFAR-10 and CIFAR-100 benchmarks. As shown in the table, our approach significantly enhances the performance of all methods on both CIFAR-10 and CIFAR-100 datasets. Notably, the maximum reductions in FPR95 on CIFAR-10 and CIFAR-100 are 66.03% (from 26.55 of Energy to 9.02 of NAP-E) and 58.71% (from 62.27 of ReAct to 25.71 NAP-R), respectively. Note that the table showcases the average performance across six OOD datasets; for complete performance details, refer to Appendix G.

## 5.2 EVALUATION ON IMAGENET

**Implementation details.** In line with recent research (Sun et al., 2021; Sun & Li, 2022; Djurisic et al., 2022), we conduct experiments with NAP on the expansive ImageNet-1k (Deng et al., 2009) dataset in this study. Four dataset subsets, with all overlapping categories with ImageNet-1k eliminated, were employed as OOD benchmarks. These OOD datasets comprise Textures (Cimpoi et al., 2014), Places365 (Zhou et al., 2017), iNaturalist (Van Horn et al., 2018), and SUN (Xiao et al., 2010). We used MobileNetV2 (Sandler et al., 2018) architecture, which pre-trained on ImageNet-1k. The architecture and parameters remain unchanged during the OOD detection stage.

**Experimental results.** Table 3 shows the comparison results of NAP with other post-hoc OOD detection methods on the ImageNet-1k benchmark. As shown in the table, our approach significantly enhances the performance of all methods on the ImageNet-1k dataset. Notably, the maximum reductions in FPR95 can be up to 45.31% (from 62.39 of Energy to 34.12 of NAP-E).

# 6 ABLATION STUDY

## 6.1 ORACLE VALIDATION OF SNR-LIKE SCORING FUNCTION DESIGN

In an "Oracle" experiment, we sought to validate the rationale behind the SNR-like scoring function design by utilizing both ID and OOD labels, although OOD data is unavailable in real-world applications. By treating maximum and mean activation values across all channels as features for each sample, we trained a MLP to perform binary classification between ID and OOD samples. The correlation analysis between MLP scores on the test set and the $S_{NAP}$ scores is detailed in Figure 4a. This approach effectively demonstrates the ability of the SNR-like score to distinguish OOD samples, thereby affirming the validity of the NAP scoring methodology.

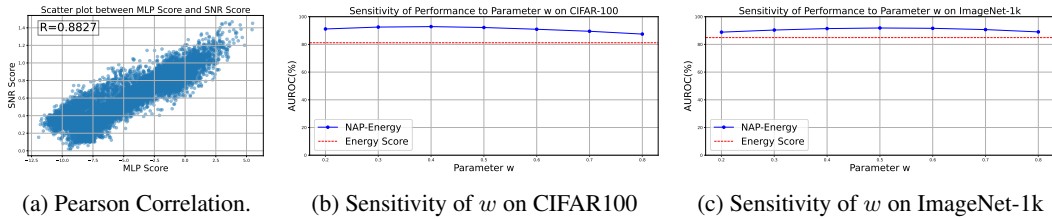

| (a) Pearson Correlation. | (b) Sensitivity of $w$ on CIFAR100 | (c) Sensitivity of $w$ on ImageNet-1k |

Figure 4: Result of validation experiment and visualization of sensitivity of the parameter $w$.

## 6.2 SENSITIVITY OF THE PARAMETER $w$

Figures 4b and 4c illustrate the sensitivity of $w$ on CIFAR (Krizhevsky et al., 2009) and ImageNet-1k (Deng et al., 2009) datasets. As shown, NAP consistently improves the baseline across the entire range of $w$ values. Additionally, the optimal $w$ search method described in Appendix H reliably finds the best w without requiring additional OOD data. Considering the simplicity of implementing NAP, its practicality remains high.

We also conducted an ablation experiment to demonstrate the impact on the results when the search for the optimal parameter $w$ is disabled. In Table 4, we directly multiplied the NAP score with the scores from other methods. It is evident that even without using the optimal hyperparameters, the effect of NAP remains significant.

## 6.3 EVALUATION OF $S_{NAP}$ IN ISOLATION

Below are the results when using $S_{NAP}$ independently on the VGG (Simonyan & Zisserman, 2015) with the ImageNet-1k dataset. Since $S_{NAP}$ does not require any hyperparameters, it is compared with MSP and Energy methods, which also do not involve hyperparameters. The results, indicating $S_{NAP}$'s strong performance on its own, are presented in the Table 5.

Table 4: Results without optimal $w$ searching. All values are percentages and are averaged over 6 OOD datasets. Note: A. = Area Under the ROC Curve; F. = False Positive Rate at 95% True Positive Rate. Methods include MSP (Hendrycks & Gimpel, 2016), Energy (Liu et al., 2020), ASH (Djurisic et al., 2022), DICE (Sun & Li, 2022), ReAct (Sun et al., 2021), and KNN (Sun et al., 2022).

| Method | | MSP | $\times S_{NAP}$ | Energy | $\times S_{NAP}$ | ASH | $\times S_{NAP}$ | DICE | $\times S_{NAP}$ | ReAct | $\times S_{NAP}$ | KNN | $\times S_{NAP}$ |
|---|---|---|---|---|---|---|---|---|---|---|---|---|---|
| **CIFAR-10** | F. ↓ | 48.69 | **19.09** | 26.55 | **9.71** | 15.05 | **11.14** | 20.83 | **11.66** | 26.45 | **9.47** | 16.12 | **10.58** |
| | A. ↑ | 92.52 | **95.11** | 94.67 | **98.00** | 96.91 | **97.48** | 95.24 | **97.47** | 94.67 | **98.11** | 96.79 | **97.74** |
| **CIFAR-100** | F. ↓ | 80.13 | **58.64** | 68.45 | **35.13** | 41.40 | **36.13** | 49.72 | **36.29** | 62.27 | **25.71** | 44.91 | **37.07** |
| | A. ↑ | 74.36 | **86.03** | 81.19 | **92.19** | 90.02 | **91.16** | 87.23 | **91.97** | 84.47 | **93.18** | 86.58 | **89.62** |
| **ImageNet-1k** | F. ↓ | 73.51 | **48.99** | 62.39 | **35.25** | 35.66 | **31.55** | 41.92 | **32.85** | 45.02 | **37.70** | 70.92 | **47.41** |
| | A. ↑ | 79.00 | **88.14** | 84.91 | **91.41** | 92.13 | **92.89** | 89.60 | **91.64** | 89.47 | **90.93** | 73.78 | **84.08** |

## 6.4 DIFFERENT DESIGN OF SCORE FUNCTION

In addition to the primary scoring function detailed in Section 4.2, we evaluate several variants in combined with the Energy score on CIFAR-10. Table 6 presents results for the following alternatives: (i) Variance: Utilizes intra-channel variance. (ii) No Square: Employs an unsquared SNR score. (iii) Minus: Leverages the differential between maximum and mean values.

Table 5: Performance on ImageNet-1K Using VGG Network. This table showcases the performance of the $S_{NAP}$ used independently, demonstrating its effective OOD detection capabilities. Comparisons are made with MSP and Energy, which similarly require no hyperparameters or training.

| Method | OOD Datasets | | | | | | | | Average | |
|---|---|---|---|---|---|---|---|---|---|---|
| | iNaturalist | | SUN | | Places | | Textures | | | |
| | FPR95 ↓ | AUROC ↑ | FPR95 ↓ | AUROC ↑ | FPR95 ↓ | AUROC ↑ | FPR95 ↓ | AUROC ↑ | FPR95 ↓ | AUROC ↑ |
| MSP | 58.70 | 86.51 | 73.60 | 79.17 | 75.74 | 78.71 | 64.26 | 82.00 | 68.07 | 81.60 |
| Energy | 51.37 | 90.30 | 57.56 | 87.55 | 64.20 | 84.83 | 44.24 | 89.98 | 54.34 | 88.17 |
| NAP | **30.20** | **94.14** | **33.48** | **91.99** | **49.40** | **86.09** | **18.05** | **96.48** | **32.78** | **92.18** |

Table 6: Ablation on score functions designs. Parentheses indicate changes from the Energy baseline.

| Method | | Energy (baseline) | Variance | No Square | Minus | $S_{NAP}$ |
|---|---|---|---|---|---|---|
| **CIFAR-10** | FPR95 ↓ | 26.55 | 13.80 (-12.75) | 12.40 (-14.15) | 12.26 (-14.29) | **9.02** (-17.53) |
| | AUROC ↑ | 94.67 | 97.08 (+2.41) | 97.52 (+2.85) | 97.51 (+2.84) | **98.15** (+3.48) |

## 6.5 COMPUTATIONAL EFFICIENCY

The computational overhead introduced by NAP is minimal. In our ImageNet-1k experiments, NAP added only 0.0817 seconds to a total runtime of 176.7155 seconds, resulting in a negligible increase of just 0.0462% in processing time.

## 7 CONCLUSION

This paper presents a novel Neural Activation Prior for OOD detection, based on the observation that in-distribution samples typically generate stronger activations in specific channel neurons compared to OOD samples. Our method offers simplicity, easy integration, and maintains in-distribution classification performance without requiring additional training or external data. Extensive experiments across various datasets and architectures demonstrate the robustness and generalizability of our approach. Specifically, our prior enriches the community's library of priors for OOD detection due to its orthogonality to previous ones, enhancing detection accuracy across a variety of existing methods. Our findings indicate that when used independently, our OOD detection method achieves performance comparable to current state-of-the-art techniques. Moreover, integrating our approach with existing methods leads to significant improvements, reducing FPR95 by up to 66.03%. These results confirm that our prior captures complementary information, highlighting its potential to advance OOD detection and contribute to enhanced reliability in machine learning systems.

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

APPENDIX

In this appendix, we provide comprehensive additional materials to supplement the main text. The contents include:

- **Broader impacts (Section A):** A discussion on the broader implications of our research.
- **Pseudo code for NAP (Section B):** The algorithmic representation of the Neural Activation Prior (NAP) methodology.
- **Why is the penultimate layer more effective for NAP? (Section C):** An exploration of the reasons behind the superior effectiveness of the penultimate layer in the context of NAP.
- **Evaluating multi-layer integration with NAP for OOD detection (Section D):** Investigation into the effects of integrating multiple layers along with NAP in detecting OOD data.
- **On transferability to other architectures (Section E):** To ascertain the versatility and robustness of NAP across different CNN architectures, we conducted extensive experiments on various backbones, including VGG, DenseNet, and ResNet.
- **Pareto frontier of ID accuracy and OOD detection performance (Section F):** Evaluation on Pareto Frontier of ID accuracy and OOD Detection Performance.
- **Full CIFAR benchmark results: enhancing methods with NAP (Section G):** Comprehensive evaluation of NAP's effectiveness in enhancing existing models, as demonstrated through detailed results on the CIFAR benchmark.
- **How to find an optimal parameter $w$? (Section H):** A guide on determining the optimal parameter $w$ for NAP.
- **Performance on Near-OOD detection (Section I):** Investigating the capability of the NAP in distinguishing between closely related datasets provides insight into its utility in nuanced OOD detection scenarios.
- **More examples of activation map visualizaiton. (Section J):** Additional visual examples showcasing the activation maps.
- **Extension to Transformer backbones. (Section K):** Following the experimental setup described in Section 5.2, we conduct experiments on the Vision Transformer(Dosovitskiy et al., 2020) (ViT-B/16) using ImageNet-1k as the ID dataset.
- **Limitations (Section L):** A critical analysis of the limitations of our approach.
- **Discussion (Section M):** A concluding section that summarizes the key findings and outlines future directions for research based on our work.
- **Licenses for existing assets (Section N):** Credits and licenses for all existing assets used in this research.

## A  BRODER IMPACTS

The proposed Neural Activation Prior for OOD detection has significant implications for the deployment of machine learning models in real-world scenarios. By enhancing the ability to detect OOD samples, our method contributes to improving the reliability and safety of AI systems, particularly in critical applications such as autonomous driving, healthcare, and security, where encountering unexpected inputs could have severe consequences.

## B  PSEUDO CODE FOR NAP

As illustrated in Algorithm 1, we present a detailed pseudo-code representation of our proposed method for OOD detection, which is integrated into the DenseNet architecture. The key modification involves the calculation of NAP score $\mathcal{S}$ within the DenseNet's processing pipeline (highlighted in green font in the algorithm), which is then followed by calculating the OOD score using the model's logits together with $\mathcal{S}$. These calculations do not alter the logits output by the model, thereby ensuring no degradation in classification accuracy for the ID dataset.

---

**Algorithm 1** OOD Detection Using Neural Activation Prior on DenseNet

---

**Require:** Image $x$, Weight $w$ (for OOD score calculation)
**Ensure:** Output logits, OOD Score
 1: Apply initial layers of DenseNet on $x$ to obtain intermediate output:
 2:    $out \leftarrow \text{conv1}(x)$
 3:    $out \leftarrow \text{trans1}(\text{block1}(out))$
 4:    $out \leftarrow \text{trans2}(\text{block2}(out))$
 5:    $out \leftarrow \text{block3}(out)$
 6:    $out \leftarrow \text{relu}(\text{bn1}(out))$
 7:    Let $\mathcal{A}$ denote this intermediate output.
 8: Compute NAP score $\mathcal{S}$ from $\mathcal{A}$:
 9:    Flatten $\mathcal{A}$ across dimensions 2 and 3.
10:    Compute Max of $\mathcal{A}$ across flattened dimensions.
11:    Compute Mean of $\mathcal{A}$ over dims 2, 3.
12:    Calculate $\mathcal{S}$ as $(\text{Max of } \mathcal{A}/(\text{Mean of } \mathcal{A} + 1))^2$.
13:    Compute the mean of $\mathcal{S}$ across dimension 1.
14: Continue with DenseNet forward pass:
15:    Apply average pooling and reshape on $\mathcal{A}$.
16:    Get logits from fully connected layer.
17: Calculate OOD Score:
18:    Compute log-sum-exp of the logits.
19:    Calculate OOD Score as $(\text{log-sum-exp}^{w}) * (\mathcal{S}^{1-w})$.
20: **return** Output logits, OOD Score

---

## C   WHY IS THE PENULTIMATE LAYER MORE EFFECTIVE FOR NAP?

We provide an extensive collection of visualizations showcasing the activations within the DenseNet architecture when applied to the CIFAR-10 (ID) dataset and Places365 (OOD) dataset. These visualizations are crucial for understanding how the network processes both ID and OOD data, revealing the distinct patterns of neural activations at various layers of the network. Our analysis focuses on four critical layers within DenseNet: (1) after the first convolutional layer, (2) before the pooling operation in the first transition block, (3) before pooling in the second transition block, and (4) before the final global pooling layer. Each layer offers four visualizations, providing a comprehensive view of the network's response to different datasets.

These detailed visualizations enhance the discussion in the main text, offering deeper insights into how NAP effectively distinguishes between ID and OOD samples within the network. As depicted in Figure 5, the first three selected layers, which focus primarily on low-level features, exhibit a less pronounced distinction between ID and OOD samples. This is likely because low-level features, such as edges and textures, are common to both ID and OOD datasets, making them less distinctive. However, the contrast between ID and OOD samples becomes more evident and stable in the fourth selected layer, located before the final global pooling layer. This layer, concentrating on high-level semantic information, captures features more unique to the ID dataset, leading to clearer separability and enhanced stability in activation values compared to earlier layers. This layer's focus on distinctive semantic features makes it particularly suitable for developing a scoring function for OOD detection. Therefore, since the penultimate layer is the most informative layer in the neural network, we utilize this layer in our method to develop our scoring function.

## D   EVALUATING MULTI-LAYER INTEGRATION WITH NAP FOR OOD DETECTION

In an additional exploration presented in the appendix, we investigate the effects of incorporating values from both the penultimate layer and earlier layers on OOD detection. Our experiments, detailed in Table 7, suggest that the integration of earlier layers with the penultimate layer, where NAP is primarily applied, does not yield significant improvements in OOD detection. This phenomenon could partly stem from the inherent limitations of earlier layers in differentiating between ID and

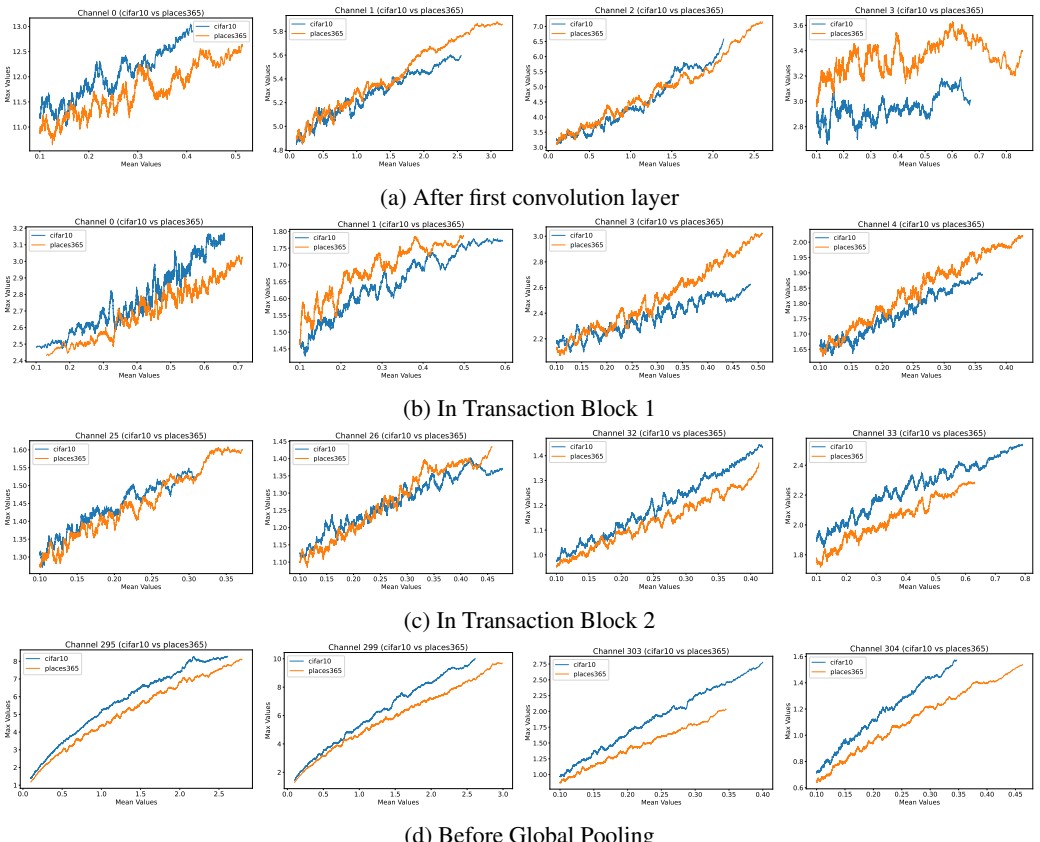

(a) After first convolution layer

(b) In Transaction Block 1

(c) In Transaction Block 2

(d) Before Global Pooling

Figure 5: **Activation distribution at different positions within the DenseNet architecture applied to CIFAR-10 and Places365 datasets.** For this analysis, four specific locations within the network were chosen: (a) after the first convolution layer, (b) just before the pooling operation in the first transition block, (c) just before the pooling operation in the second transition block, and (d) right before the final global pooling layer. Note that we only include data points with an average activation over 0.1. As shown in the figure, the first three selected layers show a less marked distinction between ID and OOD samples, while the fourth layer, preceding the final global pooling layer, demonstrates clearer separability and enhanced stability. Therefore, the fourth selected layer (the penultimate layer) is more suitable for developing a scoring function for OOD detection.

OOD data. Additionally, there is a possibility that the scoring function, specifically optimized for the penultimate layer, may not align optimally with the feature representation characteristics of the preceding layers. Considering the constraints of space in this paper, a comprehensive analysis of multi-layer integration using NAP is not presented. Nonetheless, the potential of combining multiple layers in OOD detection, especially in the context of NAP, remains an intriguing aspect for future research. We anticipate that further investigations, potentially involving the creation of new scoring functions suitable for a broader range of layers, could provide substantial contributions to the field. Thus, we propose this as an avenue for future work, aiming to stimulate further advancements within the research community.

## E ON TRANSFERABILITY TO OTHER ARCHITECTURES

In Figure 9, 10, and 11, we present detailed visualizations of the activation patterns within three distinct architectures: MobileNetV2 (Sandler et al., 2018), ResNet50 (He et al., 2016), and VGG16 (Simonyan & Zisserman, 2015). These visualizations clearly demonstrate a remarkable gap between ID samples, depicted with blue lines, and OOD samples, represented with orange lines. This distinction is evident across all three architectures, underscoring the versatility and effectiveness of the proposed

Table 7: **Experimental results of OOD detection using NAP at different layers in DenseNet on CIFAR-10 and CIFAR-100 datasets.** NAP(x) indicates the computation of OOD score at the layer 'x', where 'c1' corresponds to after the first convolutional layer, 't1' before the pooling operation in the first transition block, 't2' before pooling in the second transition block, and 'p' before the final global pooling layer. Combinations of layers, indicated by commas in NAP(..), represent the multiplication of OOD scores from respective layers. These selected layers are consistent with those used for visualizations described earlier in the paper. Notably, NAP(p) is the approach actually utilized in our paper.

| Method | CIFAR-10 | | CIFAR-100 | |
|---|---|---|---|---|
| | FPR95 | AUROC | FPR95 | AUROC |
| NAP(c1) | 83.22 | 51.99 | 84.13 | 50.34 |
| NAP(t1) | 69.10 | 50.47 | 86.82 | 54.92 |
| NAP(t2) | 56.53 | 78.44 | 88.85 | 53.08 |
| NAP(c1,t1,t2,p) | 68.33 | 58.99 | 82.66 | 56.96 |
| NAP(t1,t2,p) | 56.84 | 64.26 | 83.35 | 57.97 |
| NAP(t2,p) | 34.41 | 87.81 | 82.74 | 58.31 |
| **NAP(p)** | **26.57** | **92.45** | **54.91** | **85.86** |

NAP. The consistent separability observed in these diverse architectures confirms the adaptability and potential of NAP for broad application in different neural network models.

In order to validate the effectiveness of NAP across different Convolutional Neural Network (CNN) architectures, we conducted experiments on a variety of CNN backbones. As depicted in Table 8, our proposed NAP method significantly enhances the OOD detection performance across various CNN structures. These results underscore the adaptability of NAP to various CNN models, demonstrating its potential as a versatile tool for enhancing the reliability and accuracy of OOD detection in neural network applications.

Table 8: Results on ImageNet-1k with various backbones.

| | Energy | | NAP | | NAP-E | |
|---|---|---|---|---|---|---|
| | FPR95 | AUROC | FPR95 | AUROC | FPR95 | AUROC |
| VGG | 54.34 | 88.17 | 29.23 | 93.46 | **23.23** | **95.00** |
| DenseNet | 50.40 | 87.66 | 49.89 | 88.40 | **32.95** | **91.68** |
| ResNet | 57.47 | 87.05 | 48.77 | 82.76 | **32.12** | **92.02** |

# F PARETO FRONTIER OF ID ACCURACY AND OOD DETECTION PERFORMANCE

Some existing methods negatively affect ID accuracy; however, we have found that integrating these methods with the NAP approach can mitigate or reduce this impact, achieving a more optimal balance. The NAP approach establishes an ideal equilibrium between ID classification accuracy and OOD detection efficacy (measured in AUROC), thereby positioning it favorably on the Pareto front for superior performance. This demonstrates our method's ability to enhance OOD detection without additional costs while maintaining the model's classification performance, as illustrated in Figures 6a and 6b using the CIFAR benchmark.

# G FULL CIFAR BENCHMARK RESULTS: ENHANCING METHODS WITH NAP

This section focuses on the significant enhancements brought by the Neural Activation Prior (NAP) to existing out-of-distribution detection methods in the context of the CIFAR-10 (in Table 9) and CIFAR-100 (in Table 10) dataset. The incorporation of NAP into established methods like MSP (Hendrycks

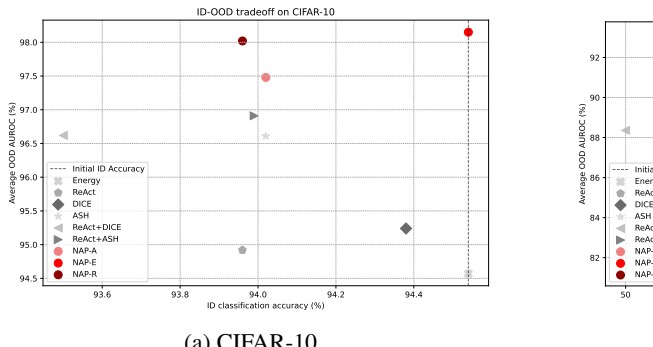
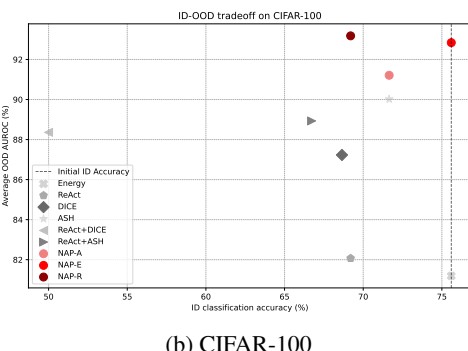

| (a) CIFAR-10 | (b) CIFAR-100 |

Figure 6: **Investigating the trade-offs between ID classification accuracy and OOD detection AUROC on CIFAR benchmarks across various methods.** All methods and experiments are implemented by us. Methods prefixed with 'NAP' are visually distinguished, highlighted in various shades of red in the figures.

& Gimpel, 2016), Energy (Liu et al., 2020), ASH (Djurisic et al., 2022), DICE (Sun & Li, 2022), SCALE (Xu et al., 2023), ReAct (Sun et al., 2021), and KNN (Sun et al., 2022), resulting in NAP-M, NAP-E, NAP-A, NAP-D, NAP-S, NAP-R, and NAP-K, respectively, showcases the potential of NAP in augmenting existing approaches. Our experimental results demonstrate that NAP-based variants consistently outperform their corresponding traditional methods across all six OOD datasets. Notably, our experimental results reveal some extremely substantial decreases in FPR95 values, indicative of the profound impact of NAP integration. For instance, on the CIFAR-100 dataset, the FPR95 value of NAP-R, compared to ReAct, dropped by 83.06% (from 83.81 to 14.19), highlighting a notable reduction in false alarms while maintaining high detection accuracy and affirming the enhanced capability of these methods in distinguishing between in-distribution and OOD samples.

Table 9: **Scoring function proposed based on NAP is compatible with and improves on existing methods on CIFAR-10 dataset.** All methods and experiments are implemented by us. All values in this table are percentages. The average over six OOD test datasets is also reported. The methods with prefix 'NAP-' (e.g., NAP-E, NAP-A, NAP-R) represent the integration of NAP with various existing methods (Energy, ASH-S, ReAct, respectively).

| Method | OOD Datasets | | | | | | | | | | | | Average | |
| | SVHN | | Textures | | iSUN | | LSUN | | LSUN-Crop | | Places365 | | | |
| | FPR95 | AUROC | FPR95 | AUROC | FPR95 | AUROC | FPR95 | AUROC | FPR95 | AUROC | FPR95 | AUROC | FPR95 | AUROC |
|---|---|---|---|---|---|---|---|---|---|---|---|---|---|---|
| MSP | 47.34 | 93.48 | 33.66 | 95.54 | 42.21 | 94.51 | 42.42 | 94.52 | 64.52 | 88.14 | 61.98 | 88.95 | 48.69 | 92.52 |
| **NAP-M** | 14.09 | 96.05 | 7.33 | 98.35 | 10.91 | 97.72 | 11.20 | 97.55 | 16.42 | 96.23 | 54.61 | 84.76 | 19.09 | 95.11 |
| Energy | 40.57 | 93.99 | 56.29 | 86.42 | 10.07 | 98.07 | 9.28 | 98.12 | 3.81 | 99.15 | 56.59 | 92.01 | 26.59 | 94.63 |
| **NAP-E** | 8.32 | 98.36 | 11.65 | 97.72 | 1.77 | 99.57 | 1.50 | 99.60 | 0.99 | 99.76 | 29.89 | 93.91 | 9.02 | 98.15 |
| ASH | 6.51 | 98.65 | 24.34 | 95.09 | 5.17 | 98.90 | 4.96 | 98.92 | 0.90 | 99.73 | 48.45 | 88.34 | 15.05 | 96.91 |
| **NAP-A** | 5.55 | 98.86 | 10.51 | 97.90 | 3.04 | 99.32 | 2.68 | 99.40 | 0.80 | 99.80 | 44.28 | 89.59 | 11.14 | 97.48 |
| DICE | 29.62 | 94.66 | 0.38 | 99.90 | 4.43 | 99.03 | 5.14 | 98.97 | 45.87 | 86.97 | 45.32 | 90.29 | 21.79 | 94.97 |
| **NAP-D** | 10.60 | 97.75 | 0.41 | 99.88 | 2.03 | 99.48 | 2.69 | 99.41 | 13.85 | 96.98 | 40.40 | 91.31 | 11.66 | 97.47 |
| SCALE | 5.93 | 98.72 | 1.33 | 99.74 | 3.75 | 99.23 | 3.73 | 99.23 | 23.78 | 94.97 | 35.05 | 91.73 | 12.26 | 97.27 |
| **NAP-S** | 4.87 | 99.01 | 0.71 | 99.82 | 2.03 | 99.54 | 2.34 | 99.51 | 11.87 | 97.71 | 33.76 | 92.43 | 9.26 | 98.00 |
| ReAct | 41.64 | 93.87 | 43.58 | 92.47 | 12.72 | 97.72 | 11.46 | 97.87 | 5.96 | 98.84 | 43.31 | 91.03 | 26.45 | 94.67 |
| **NAP-R** | 8.07 | 98.31 | 8.10 | 98.17 | 2.81 | 99.35 | 2.35 | 99.43 | 3.04 | 99.33 | 30.70 | 93.50 | 9.18 | 98.02 |
| KNN | 4.31 | 99.20 | 7.71 | 98.62 | 9.45 | 98.22 | 10.08 | 98.15 | 19.31 | 96.46 | 45.83 | 90.09 | 16.12 | 96.79 |
| **NAP-K** | 2.39 | 99.56 | 2.29 | 99.55 | 1.76 | 99.57 | 2.45 | 99.47 | 3.58 | 99.34 | 34.27 | 92.80 | 7.79 | 98.38 |

# H    HOW TO FIND AN OPTIMAL PARAMETER $w$?

When combining NAP with different OOD detection methods, the optimal weight parameter $w$ varies. To obtain the optimal parameter, we utilized a set of data transformation techniques (including Gaussian noise, shot noise, impulse noise, defocus blur, glass blur, motion blur, zoom blur, snow, frost, fog, brightness, contrast, elastic transform, pixelate, jpeg compression) to generate a corrupted dataset based on the ID dataset, serving as pseudo OOD data. Utilizing this set of OOD data, we employed a binary search method to find the optimal $w$. Through experimentation with various datasets and methods, we found that this search approach quickly identifies the optimal $w$, which

Table 10: **Scoring function proposed based on NAP is compatible with and improves on existing methods on CIFAR-100 dataset.** All methods and experiments are implemented by us. All values in this table are percentages. The average over six OOD test datasets is also reported. The methods with prefix 'NAP-' (e.g., NAP-E, NAP-A, NAP-R) represent the integration of NAP with various existing methods (Energy, ASH-S, ReAct, respectively).

| Method | OOD Datasets | | | | | | | | | | | | Average | |
| | SVHN | | Textures | | iSUN | | LSUN | | LSUN-Crop | | Places365 | | | |
| | FPR95 | AUROC | FPR95 | AUROC | FPR95 | AUROC | FPR95 | AUROC | FPR95 | AUROC | FPR95 | AUROC | FPR95 | AUROC |
|---|---|---|---|---|---|---|---|---|---|---|---|---|---|---|
| MSP | 81.70 | 75.40 | 60.49 | 85.60 | 85.24 | 69.18 | 85.99 | 70.17 | 84.79 | 71.48 | 82.55 | 74.31 | 80.13 | 74.36 |
| **NAP-M** | 35.58 | 93.32 | 15.29 | 96.94 | 66.86 | 86.62 | 57.64 | 88.98 | 27.85 | 93.93 | 86.00 | 70.89 | 48.20 | 88.45 |
| Energy | 87.46 | 81.85 | 84.15 | 71.03 | 74.54 | 78.95 | 70.65 | 80.14 | 14.72 | 97.43 | 79.20 | 77.72 | 68.45 | 81.19 |
| **NAP-E** | 19.03 | 96.40 | 21.72 | 95.47 | 33.24 | 94.15 | 43.38 | 92.11 | 2.60 | 99.38 | 75.70 | 79.54 | 32.61 | 92.84 |
| ASH | 25.02 | 95.76 | 34.02 | 92.35 | 46.67 | 91.30 | 51.33 | 90.12 | 5.52 | 98.84 | 85.86 | 71.62 | 41.40 | 90.02 |
| **NAP-A** | 17.41 | 96.72 | 22.70 | 94.99 | 38.22 | 93.34 | 43.05 | 92.17 | 5.25 | 98.94 | 85.76 | 72.08 | 35.40 | 91.32 |
| DICE | 59.25 | 88.57 | 0.91 | 99.74 | 51.63 | 89.32 | 49.48 | 89.51 | 61.42 | 77.12 | 80.29 | 77.08 | 50.50 | 86.89 |
| **NAP-D** | 23.63 | 95.28 | 1.22 | 99.68 | 33.86 | 94.25 | 28.56 | 95.02 | 24.59 | 92.04 | 82.12 | 77.13 | 32.34 | 92.23 |
| SCALE | 15.94 | 96.29 | 4.56 | 99.16 | 32.76 | 90.96 | 28.99 | 92.12 | 36.74 | 92.33 | 75.40 | 72.64 | 32.40 | 90.58 |
| **NAP-S** | 12.21 | 97.09 | 4.89 | 98.96 | 25.43 | 93.06 | 21.96 | 94.21 | 24.81 | 95.48 | 74.82 | 71.46 | 27.35 | 91.17 |
| ReAct | 83.81 | 81.41 | 77.78 | 78.95 | 65.27 | 86.55 | 60.08 | 87.88 | 25.55 | 94.92 | 82.65 | 74.04 | 62.27 | 84.47 |
| **NAP-R** | 14.19 | 96.52 | 17.22 | 96.16 | 16.72 | 96.54 | 17.16 | 96.64 | 5.73 | 98.76 | 82.54 | 74.46 | 25.71 | 93.18 |
| KNN | 16.27 | 96.65 | 28.06 | 92.69 | 58.74 | 82.09 | 52.77 | 84.55 | 26.01 | 93.53 | 87.59 | 69.94 | 44.91 | 86.58 |
| **NAP-K** | 10.26 | 97.84 | 12.24 | 97.76 | 45.56 | 91.57 | 36.45 | 93.22 | 9.84 | 98.04 | 87.42 | 70.83 | 33.63 | 91.54 |

Table 11: Optimal weight parameter $w$ for different OOD detection methods across various datasets.

| Method | CIFAR-10 | CIFAR-100 | ImageNet-1k |
|---|---|---|---|
| MSP | 0.5 | 0.3 | 0.3 |
| Energy | 0.4 | 0.4 | 0.6 |
| ASH | 0.5 | 0.6 | 0.8 |
| DICE | 0.5 | 0.6 | 0.6 |
| SCALE | 0.4 | 0.5 | 0.4 |
| ReAct | 0.4 | 0.5 | 0.8 |
| KNN | 0.8 | 0.8 | 0.6 |

generalizes well to real OOD datasets. The values of $w$ used in our experiments are summarized in the Table 11.

# I  PERFORMANCE ON NEAR-OOD DETECTION

Given the context of existing research, where CIFAR-10 is commonly used as the ID dataset and datasets such as SVHN and Texture are utilized as OOD datasets, the distinction in data distribution is markedly evident due to the difference in data sources. This conventional setup, however, does not adequately challenge the model with closely related distributions. Therefore, we embark on an experiment utilizing CIFAR-10 and CIFAR-100 as ID and OOD datasets, respectively, to explore the performance of NAP in scenarios where the data distributions are more closely aligned. This approach aims to assess the robustness of NAP in distinguishing between datasets with subtle differences in distribution yet distinct semantic features.

**Conclusion:** Our findings confirm that NAP is capable of effectively functioning in scenarios where the ID and OOD data distributions are closely related, showcasing its utility in near-OOD detection tasks. Table 12 demonstrates the effectiveness of NAP variants (NAP-E, NAP-R, and NAP-A) in comparison to baseline methods (Energy, ReAct, and ASH) for the task of near-OOD detection between CIFAR-10 and CIFAR-100 datasets.

Table 12: Result of CIFAR-10 vs. CIFAR-100

| | FPR95 | AUROC | | FPR95 | AUROC | | FPR95 | AUROC |
|---|---|---|---|---|---|---|---|---|
| Energy | 50.74 | 89.76 | ReAct | 48.77 | 90.55 | ASH | 48.74 | 89.93 |
| NAP-E | **44.38** | **90.69** | NAP-R | **42.94** | **90.66** | NAP-A | **44.92** | **90.07** |

To understand the mechanism by which NAP achieves this, it is essential to delve into how neural networks process and distinguish between different types of data. Neural network classifiers are adept at detecting various semantic features through different channels in the penultimate layer. It is this capability that NAP leverages to differentiate between ID and OOD data. NAP distinguishes between ID and OOD data based on the neural network's high response to specific semantic features of ID data. Thus, in principle, NAP is well-suited for semantic OOD detection, capable of effectively distinguishing between samples from closely related distributions but with different semantics (e.g., CIFAR-10 vs CIFAR-100, as shown in the table below). However, it is important to note that NAP is not intended for more fine-grained tasks, such as pixel-level industrial surface defect detection.

## J MORE EXAMPLES OF ACTIVATION MAP VISUALIZAITON

In this section, we present additional examples of activation map visualizations to further illustrate the challenges and phenomena discussed in our work. Specifically, we examine the penultimate layer's activation maps for both in-distribution (ID) and out-of-distribution (OOD) samples. The visualizations provide insights into how global pooling methods can obscure important distinctions between ID and OOD samples by averaging out the spatial distribution of activation values within channels.

As shown in Figure 8, each pair of images displays the activation maps for a given channel. The left image in each pair corresponds to an ID sample, while the right image corresponds to an OOD sample. Different channels in the penultimate layer are typically tuned to capture specific semantic features present in the training data. For ID samples, these features often result in high activation values in particular regions of the activation map. For example, certain regions in the left images of each pair exhibit very high activation values when the corresponding semantic features are present in the ID sample.

In contrast, OOD samples, which do not contain these specific semantic features, may still produce activation responses due to the model's unfamiliarity with such data. This can result in weak noise activations, as observed in the right images of each pair. This phenomenon highlights the difficulty faced by traditional OOD detection methods that rely on aggregated activation values, as discussed by Sun et al. (2021). These methods often fail to differentiate between ID and OOD samples because the average channel activations do not provide sufficient discriminative power.

However, the NAP score proposed in this paper addresses this issue by effectively distinguishing between ID and OOD samples based on a more nuanced analysis of activation patterns. The following visualizations exemplify the described behavior and underscore the importance of considering the distribution of activation values within channels for robust OOD detection.

## K EXTENSION TO TRANSFORMER BACKBONES

We observe that the classify (cls) token in the last block of Transformers can be effectively utilized as an analogue to the pooled activations used in CNNs for our method. Consequently, as illustrated in Figure 7, we calculate NAP score by employing the attention vectors associated with the **cls** token from the final Transformer block. This approach mirrors our methodological framework in CNNs, facilitating a coherent extension across both architectures.

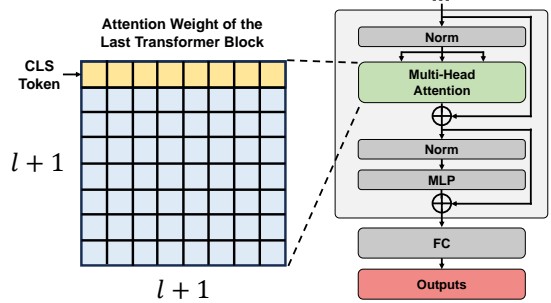

**Scoring function for Transformers.** Consistent with the NAP score used in CNNs, we calculate the mean and maximum values of the attention that the cls token has towards all other tokens. The attention vector, denoted as $A$, has a dimensionality of $(l + 1)$, where $l$ represents the sequence length. To maintain consistency with the NAP score calculation method used in CNN networks, we would typ-

Figure 7: **Illustration identifying the focus zone of NAP in classification neural networks.**

ically divide the maximum value by the mean. However, we note that the mean value of the attention vector is always $1/(l+1)$, rendering the denominator superfluous. Therefore, for simplicity, we design the NAP score function for Transformers as $S_{NAP}^{Former} = \text{Max}(A)$.

Following the experimental setup described in Section 5.2, we conduct experiments on the Vision Transformer (Dosovitskiy et al., 2020) (ViT-B/16) using ImageNet-1k as the ID dataset. Energy and MSP were selected as baseline methodologies for this analysis. The study further explores the enhancement of these baseline methods through the integration of NAP, resulting in two variants: NAP-E and NAP-M. Comparative results detailed in Table 13 demonstrate that the NAP method substantially boosts the performance on Transformer architectures beyond the baselines, affirming the utility and versatility of NAP within such contexts.

Table 13: **OOD detection results on ViT-B/16 using ImageNet-1k as ID data.** All values are percentages.

| Method | OOD Datasets | | | | | | | | Average | |
| | iNaturalist | | SUN | | Places | | Textures | | | |
| | FPR95 ↓ | AUROC ↑ | FPR95 ↓ | AUROC ↑ | FPR95 ↓ | AUROC ↑ | FPR95 ↓ | AUROC ↑ | FPR95 ↓ | AUROC ↑ |
|---|---|---|---|---|---|---|---|---|---|---|
| Energy | 64.08 | 79.24 | 72.77 | 70.25 | 74.30 | 68.44 | 58.46 | 79.30 | 67.40 | 74.31 |
| **NAP-E** | **60.97** | **80.77** | **64.05** | **77.34** | **69.34** | **73.30** | **45.04** | **86.93** | **59.85** | **79.58** |
| MSP | 51.47 | 88.16 | 66.53 | 80.93 | 68.65 | 80.38 | 60.21 | 82.99 | 61.72 | 83.12 |
| **NAP-M** | **47.09** | **88.23** | **59.45** | **82.78** | **63.38** | **80.48** | **47.70** | **87.93** | **54.40** | **84.85** |

## L    LIMITATIONS

The proposed method relies on the neural network's ability to effectively learn specific semantic features of the ID dataset in the penultimate layer, and the assumption that OOD samples do not possess these features. If OOD samples exhibit similar semantic features or if the neural network is not well-trained, the effectiveness of the proposed method may be compromised.

## M    DISCUSSION

Based on the ultra effectiveness of our prior, shown in the visualization results presented in Figure 1 in the main text and Figure 9, 10, and 11 in this appendix, we are confident that more effective scoring functions exist. Due to space limitations, the primary focus and contribution of this paper is to introduce this prior and and validate its effectiveness by proposing a new score function. We leave the development of optimal scoring functions based on our prior for future work, aiming to further contribute to the community. We also hope this paper could encourage the researchers in this community to build upon our work and advance the field.

## N    LICENSES FOR EXISTING ASSETS

We use several datasets and external code libraries in our research. To maintain anonymity, explicit references and detailed license information are not included in the submitted code. However, all creators and original owners of these assets are properly credited, and their licenses and terms of use are respected. The datasets used include CIFAR-10 (Krizhevsky et al., 2009), CIFAR-100 (Krizhevsky et al., 2009), ImageNet-1k (Deng et al., 2009), CIFAR-10-C (Hendrycks & Dietterich, 2018), CIFAR-100-C (Hendrycks & Dietterich, 2018), ImageNet-1k-C (Hendrycks & Dietterich, 2018), SVHN (Netzer et al., 2011), iSUN (Xu et al., 2015), LSUN (Yu et al., 2015), LSUN-crop (Yu et al., 2015), iNaturalist (Van Horn et al., 2018), SUN (Xiao et al., 2010), Places (Zhou et al., 2017), Textures (Cimpoi et al., 2014), and Places365 (Zhou et al., 2017). We also utilize DenseNet (Huang et al., 2017), ResNet (He et al., 2016), VGG (Simonyan & Zisserman, 2015), and ViT (Dosovitskiy et al., 2020) architectures in our experiments. Our codebase for Neural Activation Prior (NAP) includes original code under the MIT License, with additional components from other sources. External code sources include ASH (Djurisic et al., 2022), DICE (Sun & Li, 2022), and (Sun et al., 2022) under the MIT License.

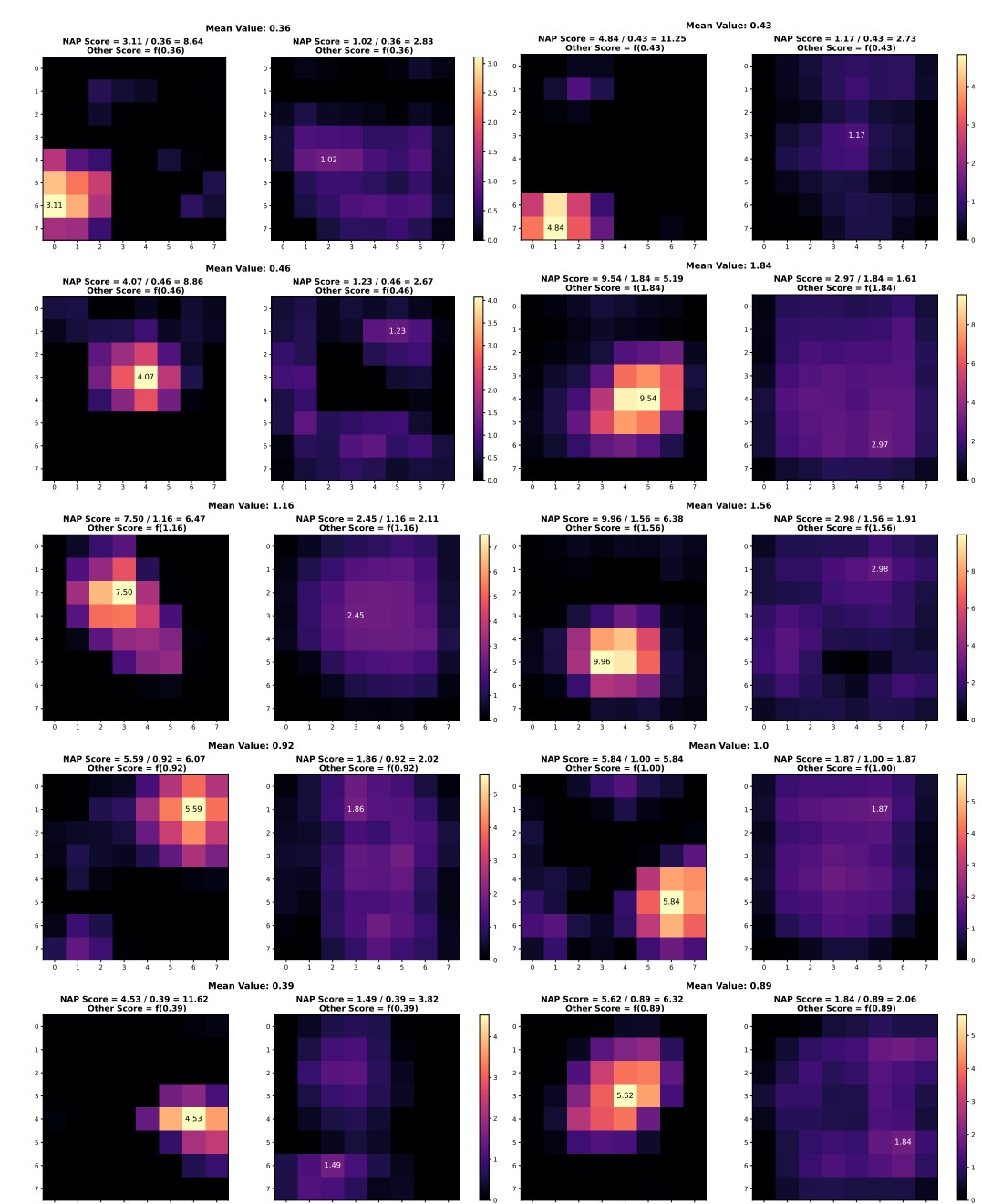

Figure 8: **Global pooling disregards the distribution of activation values within channels, making it challenging to differentiate between ID and OOD samples.** Each pair of images in this figure illustrates the activation maps of the penultimate layer for in-distribution (ID) samples (left) and out-of-distribution (OOD) samples (right) within the same channel. In this layer, different channels typically focus on distinct semantic features. When specific features are present in the image, such as certain regions in the left images of each pair, these areas exhibit very high activation values. Although OOD samples lack these specific features, the model's unfamiliarity with OOD data can lead to unpredictable activations, potentially resulting in weak noise activations (right images). This phenomenon is discussed in detail by Sun et al. (2021). Existing methods often rely on aggregating activation values for OOD detection. Consequently, the average channel activations of ID and OOD samples are not discriminative, making it difficult for existing methods to distinguish between them. However, the NAP score proposed in this paper effectively differentiates them.

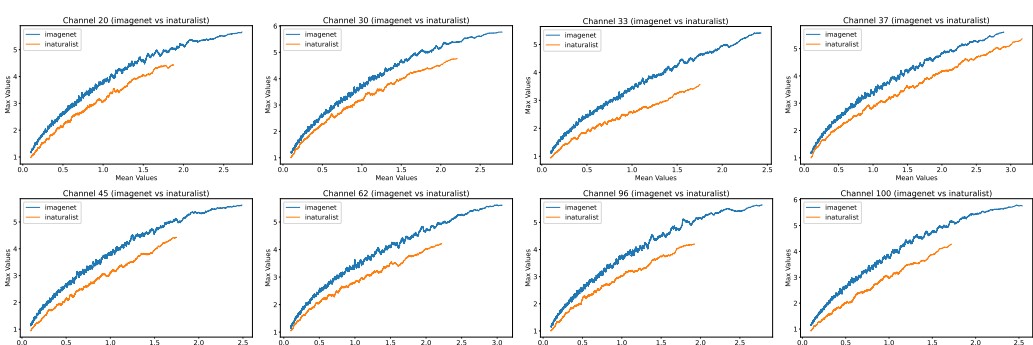

Figure 9: **Activation distribution at penultimate layer before global pooling operation within the MobileNetV2 architecture Sandler et al. (2018) applied to ImageNet-1k and iNaturalist datasets Van Horn et al. (2018).** We only include data points with an average activation over $0.1$. The figures show that our Neural Activation Prior (NAP) method is also effective in MobileNetV2, proving that NAP can be applied to different architectures.

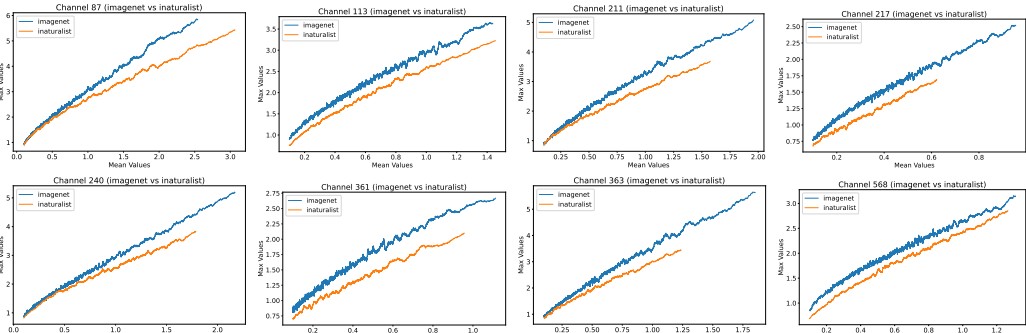

Figure 10: **Activation distribution at penultimate layer before global pooling operation within the ResNet50 He et al. (2016) architecture applied to ImageNet-1k and iNaturalist datasets Van Horn et al. (2018).** We only include data points with an average activation over $0.1$. The figures show that our Neural Activation Prior (NAP) method is also effective in ResNet50, proving that NAP can be applied to different architectures.

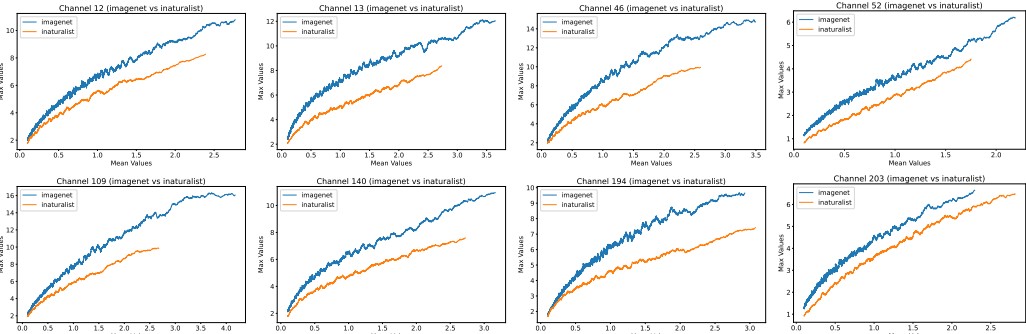

Figure 11: **Activation distribution at penultimate layer before global pooling operation within the VGG16 Simonyan & Zisserman (2015) architecture applied to ImageNet-1k and iNaturalist datasets Van Horn et al. (2018).** We only include data points with an average activation over $0.1$. The figures show that our Neural Activation Prior (NAP) method is also effective in VGG16, proving that NAP can be applied to different architectures.

