# OpenReview forum: "Out-of-Distribution Detection using Neural Activation Prior"
_ICLR.cc/2025/Conference — ICLR 2025 Conference Withdrawn Submission_

### Official Review · Reviewer_oDfM · 2024-10-31

**Soundness:** 2
**Presentation:** 3
**Contribution:** 3
**Rating:** 5
**Confidence:** 5

**Summary:**

This paper introduces a metric to out-of-distribution (OOD) detection called Neural Activation Prior (NAP). The primary goal of this work is to develop a simple yet effective plug-and-play OOD detection method that leverages intra-channel activation patterns of neural networks. The proposed NAP exploits the observation that certain neurons in channels preceding the global pooling layer have significantly higher activation probabilities for in-distribution (ID) samples compared to OOD samples. This is based on the fact that these channels are tuned to detect specific patterns learned during training on ID data.

NAP characterizes this difference by using the ratio of the maximum activation value to the average activation value within a channel, which is conceptually inspired by the Signal-to-Noise Ratio (SNR). This scoring function, named SNAP, is designed to distinguish between ID and OOD samples effectively. Importantly, NAP does not require additional training, extra data, or any significant computational overhead, which makes it broadly applicable across various architectures.

The paper also demonstrates how NAP can be combined with other existing OOD detection methods, leveraging its orthogonality to existing priors. Extensive experiments on benchmark datasets like CIFAR-10, CIFAR-100, and ImageNet-1k show that the integration of NAP with existing methods significantly enhances performance, reducing false positive rates and improving detection accuracy.

**Strengths:**

1. Unlike existing OOD methods that often rely on output-level statistics or require additional training phases, NAP leverages intra-channel activation statistics to differentiate between in-distribution (ID) and OOD samples. This novel application of intra-channel activation patterns, specifically using the ratio of maximum to mean activations inspired by Signal-to-Noise Ratio (SNR), is a creative approach to OOD detection. It broadens the range of methods used for OOD detection by focusing on internal network activations rather than just final outputs.

2. The plug-and-play aspect of NAP is original and highly practical. Unlike methods that require substantial retraining or hyperparameter tuning, NAP can be seamlessly integrated into existing models without modifications. This flexibility makes it an innovative solution for practical OOD scenarios where resources or retraining options are limited.

3. The claim that NAP is orthogonal to existing OOD detection methods is notable, allowing for a complementary integration of NAP with state-of-the-art methods such as ReAct, MSP, and Energy. This adaptability is an original feature that enhances the utility of NAP, positioning it as a versatile tool rather than a competing alternative.

**Weaknesses:**

1. On line 330, page 7, the paper states that the mean value represents noise strength, but there is insufficient explanation or justification for why this is the case. The connection between the mean activation value and noise strength, as well as its relevance to OOD detection, remains unclear. Why is the mean activation value considered to represent "noise strength"? The definition of "noise strength" is ambiguous, and the connection between activation mean and noise is not intuitive. A more rigorous explanation is needed to substantiate this assumption. A clear definition of what they mean by "noise strength" in this context is required.


2. The paper does not clearly explain why using noise strength is a reasonable criterion for OOD detection. Why should the mean (interpreted as noise strength) be an effective discriminator between ID and OOD? The rationale behind this choice is not well articulated. Provide intuition or evidence for why noise strength should differ between ID and OOD samples.

3. In the current era of large language models (LLMs) and multimodal foundational models, the importance of OOD detection is somewhat questioned, especially given that the proposed method is evaluated solely on natural image datasets.

4.  The strengths of Neural Activation Prior (NAP) compared to other established OOD detection metrics, such as ReAct, ASH, Energy, and MSP, are not made clear. The claim that NAP is "orthogonal" to these metrics is not adequately explained, and it remains unclear why combining NAP with other methods should improve performance. Provide a clear comparison of NAP's strengths and weaknesses relative to other established methods, perhaps in the form of a table or diagram

5. The paper lacks a detailed theoretical analysis of the Neural Activation Prior (NAP) concept. While the empirical results show promise, there is little discussion on why intra-channel activation patterns are effective for distinguishing in-distribution (ID) from out-of-distribution (OOD) samples from a theoretical perspective.

6. The scalability of NAP is not adequately addressed. Given that modern machine learning models often involve large architectures and massive datasets, it is important to understand how well the proposed method scales in terms of computational overhead and practicality.

**Questions:**

1. Can you provide a formal explanation for why the mean value represents "noise strength"? How is this definition consistent with existing theories or practices regarding noise in neural network activations?

2. How does your method complement or add value to the capabilities of large foundational models, which already exhibit strong performance in handling diverse data distributions? What are the specific roles of NAP in this context?

3. What are the unique strengths of NAP compared to other OOD metrics such as ReAct, ASH, Energy, and MSP? Why do you believe NAP is orthogonal to these methods?

4. Is there a theoretical explanation or hypothesis for why intra-channel activation patterns can distinguish ID from OOD effectively? Are there specific properties of CNN architectures that make these patterns reliable?

5. How does NAP scale to larger datasets and more complex architectures (e.g., ResNet-101 or Vision Transformers)? Does the calculation of intra-channel activation ratios introduce significant computational costs?

6. How does NAP perform with non-convolutional architectures, such as Vision Transformers (ViTs) or other attention-based models? Do intra-channel activation patterns still hold meaningful information in these architectures?

---

### Official Review · Reviewer_YtZR · 2024-10-31

**Soundness:** 4
**Presentation:** 4
**Contribution:** 4
**Rating:** 8
**Confidence:** 5

**Summary:**

The paper introduces Neural Activation Prior (NAP), a novel method for detecting out-of-distribution (OOD) data in machine learning. NAP leverages the activation patterns of neurons in neural networks to distinguish between in-distribution and OOD samples. The authors conduct experiments on various datasets, demonstrating NAP's effectiveness compared to existing techniques. Additionally, the paper discusses broader impacts and the versatility of NAP across different architectures.

**Strengths:**

The strengths of the Neural Activation Prior (NAP) include its ability to effectively leverage intra-channel activation patterns, enhancing OOD detection without requiring additional training or external data. NAP is a plug-and-play solution that integrates seamlessly with existing methods, improving their performance significantly. The empirical validation across multiple datasets showcases its robustness and versatility. Additionally, NAP contributes a unique perspective to the OOD detection field, enriching the community's toolkit for addressing distribution shifts. In summary, excellent results and no need for additional data or models. Very simple approach that can be implemented easily.

**Weaknesses:**

One weakness of the Neural Activation Prior (NAP) is its limited applicability to fine-grained tasks, such as pixel-level defect detection, where more detailed analysis is required. The method primarily focuses on the penultimate layer, potentially overlooking valuable information from earlier layers in the network. Additionally, while NAP enhances existing methods, it may not address all types of OOD scenarios, particularly those with significant semantic differences. Lastly, the reliance on specific activation patterns may lead to challenges in generalization across diverse datasets and architectures.

**Questions:**

No questions.

---

### Official Review · Reviewer_wTF6 · 2024-11-01

**Soundness:** 2
**Presentation:** 2
**Contribution:** 2
**Rating:** 3
**Confidence:** 4

**Summary:**

Out-of-distribution (OOD) detection is essential for deploying machine learning models in real-world scenarios, as it addresses the challenges posed by unseen data. This paper introduces a novel Neural Activation Prior (NAP) for OOD detection, based on the observation that fully trained neural networks exhibit a higher probability of strong neuron activation in response to in-distribution (ID) samples compared to OOD samples. The authors propose a scoring function that leverages this prior to enhance OOD detection without degrading ID classification performance or requiring additional training data. Their method uniquely utilizes intra-channel activation patterns, making it compatible with existing approaches and allowing for effective integration. Through comprehensive experiments, including an oracle validation, the authors demonstrate the effectiveness of their method, which significantly improves performance when combined with other OOD detection techniques.

**Strengths:**

The paper is well-written and easy to understand. This paper designs a method called NAP, which utilizes visualization to demonstrate its motivation and validates the effectiveness of the approach through extensive experiments.

**Weaknesses:**

At its core, this work is no different from Ash; both observe the most prominent activations in the feature maps. In this regard, the motivation can be considered overlapping with Ash, and the originality of the paper is not as high as claimed in the text. Furthermore, the observations regarding feature activations in this paper are consistent with the phenomena described in [1], which further diminishes the novelty of this observation.

Additionally, the score presented in this work is extremely unstable and essentially overfits to datasets with object-centric characteristics. In other words, it is designed with the assumption that the known ID datasets are of types like ImageNet and CIFAR. First, it imposes no constraints on the size of ID objects; the maximum value divided by the average value means that the magnitude of this result heavily depends on the pixel area of the ID semantics in the image. As shown in Figure 1(a), if the object were larger, resulting in high activations covering almost the entire image, the final score would actually be lower. This suggests that the effectiveness of the method when using ImageNet as the ID dataset may be coincidental. It would be worthwhile for the authors to test their method using Textures as the ID dataset to see how it performs.

Moreover, the method heavily relies on the semantic complexity within the images. Theoretically, the more semantic elements present in an image, the more likely it is to activate certain neurons' responses. Additionally, even within the same ID dataset, there can be significant differences in semantic complexity between different categories. The method tends to output lower scores for categories with sparse semantics and higher scores for those with complex semantics. It would be beneficial to discuss this issue further and to experiment with using datasets with sparse semantics as ID and more complex datasets as OOD to observe the effects.

Furthermore, there are concerns regarding the design of the Max operation. It is well-known that Max is particularly unstable. The purpose of this OOD detection algorithm is to address the issue of model overconfidence. If there is an OOD sample that closely resembles an ID sample (i.e., in a near-OOD setting), it may still conform to certain ID patterns in some channels, leading to a high score. This raises the question of whether the near-OOD experiments conducted are comprehensive enough. It would be beneficial to use larger datasets and to visualize and compare the activation heatmaps of both ID and OOD samples.

On another note, the issue is not limited to near-OOD samples; there is an even greater concern regarding spurious correlations. By using Max, the influence of spurious correlation features is amplified. It would be important for the authors to provide an explanation for this and to present experimental results on datasets with spurious correlations.

Lastly, the article appears to be based on an empirical modeling approach, lacking a reliable theoretical framework to explain its findings. As previously mentioned, there are numerous datasets and activation patterns that may not align with the assumptions made in this paper. The authors should provide a more detailed modeling and analysis to strengthen their claims.

Additionally, it is worth noting that the results reported in Table 3 for ReAct, ASH, and SCALE are significantly lower than those reported in their original papers. A similar issue arises in Table 13, where the results differ substantially from those reported in [2]. This raises concerns about the potential lowering of baseline performance in the current study.

[1] Pei, Sen, et al. "End-to-End Out-of-distribution Detection with Self-supervised Sampling." ICLR, 2024.

[2] Stanislav Fort, Jie Ren, and Balaji Lakshminarayanan. Exploring the limits of out-of-distribution
detection. In NeurIPS, 2021.

**Questions:**

see Cons

---

### Official Review · Reviewer_McuT · 2024-11-02

**Soundness:** 2
**Presentation:** 3
**Contribution:** 2
**Rating:** 3
**Confidence:** 4

**Summary:**

This paper proposes Neural Activation Prior (NAP) for OOD detection. The authors find that for a channel before the pooling layer of a fully
trained neural network, the probability of a few neurons being activated with a large response by an ID sample is significantly higher than that by an OOD sample. Based on this, the authors propose a scoring function that utilizes the ratio of maximal to averaged activation values within a channel.

**Strengths:**

* Experiments have been extensively conducted across a set of diverse datasets.
* The computational overhead introduced by the proposed NAP is minimal.

**Weaknesses:**

* The empirical performance is not satisfactory. While NAP can provide a large boost for weak baselines, for strong baselines such as ASH, the boost provided by adding NAP scores is limited. Besides, when NAP is used alone, it does not perform as well as many previous methods. For example, in Table 3, the AUROC of NAP is 89.19, which is not as good as ASH,DICE,SCALE and ReAct.
* The proposed method is not very novel. It is well known that a trained neural networks produce higher activation values for ID samples than for most OOD samples. This is used as a baseline work for OOD detection: maximum softmax probability, MSP, which uses the maximum activation value of the last layer as a scoring function. This paper seems to simply restate this phenomenon in the penultimate layer.
* Using mean value as noise doesn't seem reasonable. I understand the authors' intention to remove the influence of activation values from non-pattern regions(background regions), however the effect of the pattern regions on the mean value is greater compared to the non-patternl regions, which can also be seen in Figure 1 b, where the mean of the ID samples is higher than that of the OOD samples.
* Putting aside the concept of signal-to-noise ratio,  why do the authors use the maximum activation value divided by the mean activation value? From Figure 1 I see that both the maximum activation value and the mean activation value of the ID samples are higher than the OOD samples.  Wouldn't divided by the mean activation value weaken the proposed scoring function? What is the performance of using only the maximum activation value?

**Questions:**

see weakness.

---

### Note · Authors · 2024-11-12

I have read and agree with the venue's withdrawal policy on behalf of myself and my co-authors.